

# Holocene temperatures in southwestern Greenland controlled by topography, ice sheet proximity and oceanic conditions

Sudip Acharya[1, *], Allison A. Cluett[1,2], Amy L. Grogan[1], Jason P. Briner[1], Isla S. Castañeda[3], Elizabeth K. Thomas[1]

[1] Department of Geology, State University of New York at Buffalo, Buffalo, NY, USA
[2] University of California, Santa Cruz and NOAA Southwest Fisheries Science Center
[3] University of Massachusetts Amherst, Amherst, MA, USA

*Correspondence to*: Sudip Acharya (sudipach@buffalo.edu)

**Abstract.** The Holocene thermal maximum (HTM), a period during the early and middle-Holocene when Greenland likely experienced warmer than pre-industrial climate, provides an ideal opportunity to test the sensitivity of the Greenland Ice Sheet to prolonged warmer-than-preindustrial conditions. However, available climate reconstructions from the region provide a controversial picture of the HTM—several reconstructions show an earlier HTM between the early- to middle-Holocene, while others show a delayed HTM between the middle to late Holocene. This discrepancy may be due to either the seasonal sensitivity of the proxies or to spatio-temporal climate variations. Here we generate five new Holocene branched glycerol dialkyl glycerol tetraether (brGDGT)-inferred ice-free season lake water temperature timeseries from lakes along a latitudinal transect in southwestern Greenland, yielding a total of seven Holocene brGDGT timeseries in this region. Lake model simulations suggest minimal intra-lake variation in both the seasonal production window of brGDGTs and the sensitivity of studied lakes to air temperature changes, suggesting regional climate as a primary mechanism influencing these timeseries. Five of the brGDGT timeseries suggest a thermal maximum between approx. 7 and 5 ka, following the peak summer solar insolation and in agreement with many regional reconstructions. A coastal site that is influenced by ocean-atmosphere heat exchange experienced a thermal maximum between approx. 5 to 3 ka, coinciding with nearby sea surface temperature reconstructions. A site far from both the coast and the Greenland Ice Sheet suggests peak warmth in the early Holocene. This suggests that local variations in temperature, influenced by the proximity to the ice sheet and ocean, caused the discrepancies in the Holocene temperature reconstructions in proxy timeseries in southwestern Greenland. Further investigations quantifying seasonal sensitivity and local effects (e.g., site-specific systematics, and proximity to ice sheet and ocean) may reveal similarities among proxy timeseries.

## 1 Introduction

Arctic warming has profound and far-reaching impacts on the region's environment (Rantanen et al., 2022; Thoman et al., 2023). For example, the Greenland Ice sheet has been melting at an increasing rate, causing sea level rise (Dutton et al., 2015;



Frederikse et al., 2020). Ocean circulation and biogeochemical cycles have also undergone significant changes (Vihma et al., 2016; Wrona et al., 2016). Nutrient input into lakes have increased due to permafrost thawing (Reyes and Lougheed, 2015), leading to summer stratification and enhanced microbial respiration (Antoniades et al., 2024; Jane et al., 2023; Klanten et al.,
2023; Prowse et al., 2011). These changes raise serious concerns about the future of the Arctic environment as global warming continues. However, available climate model projections show inter-model differences (IPCC, 2021; Seneviratne and Hauser, 2020), making it difficult to accurately predict the future. To improve our understanding of the sensitivity of the Artic system, ice sheets, and the water cycle to future climate changes and to improve climate model capability, a long-term perspective on past climate changes and causal mechanisms is required (Axford et al., 2021; Briner et al., 2016).

The Holocene epoch (11.7 thousand years before present [ka] to 0 ka) on Greenland is characterized by millennia of relatively warm climate during the early to middle Holocene, likely warmer than preindustrial, and subsequent gradual cooling until 1850 CE, generally tracking boreal summer insolation (Axford et al., 2021; Briner et al., 2016; Kaufman et al., 2020). During the early and middle Holocene, the region around Greenland experienced major environmental changes: mountain glaciers were either smaller or entirely absent (Larocca et al., 2020; Larocca and Axford, 2022; Larsen et al., 2017; Schweinsberg et
al., 2017, 2019), and the Greenland Ice Sheet was smaller compared to its current extent (Briner et al., 2020; Larsen et al., 2015; Leger et al., 2024), potentially causing changes in atmospheric circulation (Masson-Delmotte et al., 2005; Thomas et al., 2016, 2023). However, the patterns of Holocene climate change in Greenland are still poorly resolved. This is in part due to site-specific, proxy-to-proxy, and proxy-model disagreement (Axford et al., 2021; Briner et al., 2016; Liu et al., 2014; Martin et al., 2024). Therefore, quantifying temperature variations throughout Greenland during this epoch is of particular
interest.

The Southwestern Greenland Ice Sheet (defined herein the region from Nugaatsiaq to Cape Farewell) is highly sensitive to Holocene climate and environmental perturbations, due to proximity to Atlantic Ocean deepwater formation, with attendant variability in heat and moisture transport (Downs et al., 2020; Lazier et al., 2002; Lesnek et al., 2020; Yang et al., 2016; Young et al., 2020). Holocene temperature timeseries from the ice-free areas in this region are mainly derived from lacustrine proxies,
including pollen, chironomids, and biomarkers (Axford et al., 2021; Briner et al., 2016; Gajewski, 2015; Kaufman, 2004). Although most of the existing time series show gradual cooling throughout the late Holocene, they disagree on the timing and magnitude of the temperature maximum, with evidence for an earlier and larger temperature maximum farther north, but with differences between archives and proxies (Axford et al., 2021; Briner et al., 2016; Gajewski, 2015; Kaufman, 2004; Larocca and Axford, 2022).

Reconstructions of Greenland Ice Sheet margin positions between Ilulissat and Kangerlussuaq suggest rapid retreat and warmest summers between ~10.4 and 9.1 ka (Lesnek et al., 2020), a time period not covered by many other terrestrial archives. An annual temperature reconstruction based on argon and nitrogen isotopes from the Greenland Ice Sheet Project (GISP2) ice core reveals a Holocene thermal maximum (HTM) at ~7.9 ka which is temperature ~2.9 ± 1.4 °C warmer than the recent





decades (Kobashi et al., 2017) (Fig. 1; point 6). Chironomid-inferred July air temperatures from North Lake near Ilulissat (Fig.
1, point 10) shows the warmest period of the past 7 ka between ~6 to 4 ka, with temperatures ~2 to 3 °C higher than preindustrial
(Axford et al., 2013). A branched glycerol dialkyl glycerol tetraether (brGDGT)-inferred summer lake water temperature time
series from Lake Gus between Kangerlussuaq and Sisimiut reveals maximum temperature of the past 9 ka from 9 to 6 ka
(Cluett et al., 2023) (Fig. 1, point 11), whereas brGDGTs from Lake 578 in southern Greenland reveals the warmest period in
the past 11 ka from ~7.5 to 4.5 ka, with ice-free season temperature ~4 to 6 °C higher than preindustrial in both timeseries
(Schneider et al., 2024) (Fig. 1, point 14). Alpine glaciers from southern Greenland were at their smallest extent between ~7.3-
7.1 ka, implying warmest temperature in the past ~10.5 ka, with air temperature only ~1 to 2 °C above present (Larocca et al.,
2020) (Fig. 1; point 15, 18). Pollen based reconstructions from Lake Qipisarqo suggest that the thermal maximum of the past
8.5 ka occurred between ~7 and 5 ka, with summer air temperatures up to ~5 °C warmer than pre-industrial (Frechette and de
Vernal, 2009) (Fig. 1; point 12). In contrast, nearby pollen-based reconstructions from Spongilla Sø and Comarum Sø imply
warmest summers of the past 10.5 ka occurred after 5.2 ka, with magnitude of ~1 to 2 °C warmer than present (Fredskild,
1973; Gajewski, 2015) (Fig. 1; point 14 and 16). Additionally, marine reconstructions indicate the HTM between ~6 and 2 ka,
with summer sea surface temperatures (SST) up to ~5 °C warmer than present (Axford et al., 2021; Hansen et al., 2020;
Ouellet-Bernier et al., 2014).

Several mechanisms have been called upon to explain the different timing and magnitude of peak warmth: differing seasonal
sensitivity of proxies, ice sheet meltwater influence causing delayed warmth in marine archives, and vegetation lags causing
delays in pollen records (Axford et al., 2021; Briner et al., 2016). Some of these mechanisms can be tested by generating time
series from the same proxy in similar archives along a spatial gradient, thus avoiding possible differences caused by the
archive/proxy.

Here we developed new brGDGT-inferred ice-free season lake water temperature timeseries from five lakes along a latitudinal
transect from 64 to 69° N (Fig. 1) in southwestern Greenland, spanning the past 9 ka. BrGDGTs are membrane-spanning lipids
of bacteria (Sinninghe Damsté et al., 2000) and they are well preserved in lake sediments (Castañeda and Schouten, 2011;
Schouten et al., 2013). BrGDGTs distributions are primarily influenced by temperature (Raberg et al., 2021) and therefore this
proxy has shown promise as a quantitative high-latitude terrestrial paleotemperature proxy (e.g., Cluett et al., 2023; Lindberg
et al., 2022; Schneider et al., 2024; Thomas et al., 2018). However, the influence of site-specific factors such as sub-oxic
conditions (van Bree et al., 2020; Weber et al., 2018; Wu et al., 2021), lake ice cover duration (Shanahan et al., 2013), and
microbial ecology (Liang et al., 2024) should also be taken into consideration for robust temperature reconstructions. To
investigate the influence of site-specific variations such as mixing regimes that may cause sub-oxic conditions, duration of ice
cover, and sensitivity of the lake system, we ran the lake energy, hydrologic, and isotopic mass balance model developed by
(Dee et al., 2018; Hostetler and Bartlein, 1990) and updated by (Morrill et al., 2019) for each lake under modern and perturbed





climate scenarios. Overall, we aim to investigate (1) Holocene temperature evolution in southwestern Greenland, and (2) Possible causes of temporal differences in temperature maxima.

## 2 Materials and methods

### 2.1 Study area and coring

We studied five lakes in southwestern Greenland with informal names: Pluto, N3, Rosaea, Marshall, and Bullet (Fig. 1). None
of the lakes currently receive ice-sheet meltwater. Please refer to Table 1 for detailed geographical and morphological features of each lake. Lakes Pluto (69.109° N, 51.032° W, 190 m asl; Fig. 1; point 1) and N3 (68.636° N, 50.981° W, 59 m asl; Fig. 1; point 2) are situated south of Ilulissat, western Greenland. Lake Pluto has a surface area of 0.7 km², maximum depth of 4 m, and catchment area of 1.21 km². Lake N3 is situated 50 km south of Lake Pluto and has a surface area of 0.09 km², a maximum depth of 16 m, and a catchment area of 4.7 km². Watershed vegetation consists of dwarf shrub heath, dominated by *Salix* sp.
and *Betula* sp. (Bennike, 2000). During the field expedition in July 2009 (Pluto Lake) and late July 2010 (N3), both lakes had small inflowing streams and active channelized outflows. For more details see Thomas et al., (2016; 2020).

**Table 1: Geographical and morphological characteristics including site ID in Fig. 1, latitude (lat), longitude (lon), elevation (m asl), catchment area (km²), maximum depth (m), surface area (km²), and closest estimated distances from Ice Sheet and Ocean (km²) of**
**study lakes (Pluto, N3, Rosaea, Marshall, and Bullet), Lake Gus (Cluett et al., 2023) and Lake 578 (Schneider et al., 2024).**

| Site ID. in Fig 1. | Lake name | Lat (°) | Lon (°) | Elevation (m asl) | Catchment area (km²) | Maximum depth (m) | Surface area (km²) | Distance from modern ice sheet (km) | Distance from ocean (km) |
|---|---|---|---|---|---|---|---|---|---|
| 1 | Pluto | 69.109 | -51.032 | 190 | 1.21 | 4 | 0.79 | 36 | 2 |
| 2 | N3 | 68.636 | -50.981 | 59 | 1.72 | 16 | 0.94 | 14 | 2 |
| 11 | Gus | 67.032 | -52.427 | 300 | 4.61 | 6 | 0.92 | 100 | 65 |
| 3 | Rosaea | 66.982 | -53.718 | 228 | 1.03 | 9 | 0.16 | 160 | 3 |
| 4 | Marshall | 64.464 | -49.431 | 862 | 0.56 | 8 | 0.29 | <0.5 | 150 |
| 5 | Bullet | 63.982 | -49.537 | 944 | 0.30 | 8 | 0.25 | 1.5 | 100 |
| 13 | 578 | 61.080 | 45.609 | 155 | 0.89 | 16 | 0.64 | 35 | 76 |

Lake Rosaea (66.982° N, 53.718° W; Fig. 1; point 3) is situated approx. 5 km north of Sisimiut on western Greenland and 3 km from the coastline with Davis Strait. The lake has a surface area of 0.02 km², maximum depth of 9 m, and a catchment area of 1.03 km² that consists of shrubs including *S. glauca* and *S. herbacea* on hill slopes while *E. nigrum*, *Betula nana*, and



*Rhododendron tomentosum* are present on low-lying ground. Patches of snowpack were observed around the lake and at the higher elevations during the field expedition on July 18–21, 2018. Also, during that summer, the lake was fed by multiple active channelized inflow streams, and it had an outflow stream.

Marshall (64.464° N, 49.431° W, 862 m asl; Fig. 1; point 4) and Bullet (63.982° N, 49.537° W, 944 m asl; Fig. 1; point 5) lakes are situated inland of Nuuk in southwestern Greenland. Marshall Lake has a surface area of 0.02 km², and maximum water depth of 8 m. The lake is <2 km outboard of the modern ice sheet margin and received ice-sheet meltwater briefly during the historical maximum extent. The lake has a catchment area 0.56 km². Bullet Lake is located approx. 60 km south from Marshall Lake. It has a surface area of 0.025 km², and maximum water depth of 8 m. The lake has a catchment area of 0.30 km² with vegetation including *Vaccinium uliginosum*, *Empetrum nigrum*, *Salix herbacea*, and *Sphagnum* sp. Both Marshall and Bullet lakes catchments include shrubs and grasses on hillslopes and *Eriophorum scheuchzeri,* and *Sphagnum* sp*.* around the shoreline. Both Marshall and Bullet lakes do not have channelized inflow and outflow streams.

## 2.2 Chronology

[14]C-ages and age-depth models for the sediment cores from lakes N3 and Pluto are published by Thomas et al. (2016) and Thomas et al. (2020). Here, we updated the age-depth models for the lakes employing the IntCal20 calibration curve (supplementary fig. 1–2). Additionally, we developed age-depth models for Marshall, Bullet and Rosaea lakes (supplementary fig. 3–5). The age-depth model for Lake Marshall is derived from [14]C ages from seven aquatic moss macrofossils, Bullet and Rosaea Lakes derived from nine aquatic macrofossils (supplementary table 1–3). All the age-depth models were calculated using rbacon within the GeoChronR package (Blaauw and Christen, 2011; McKay et al., 2021).



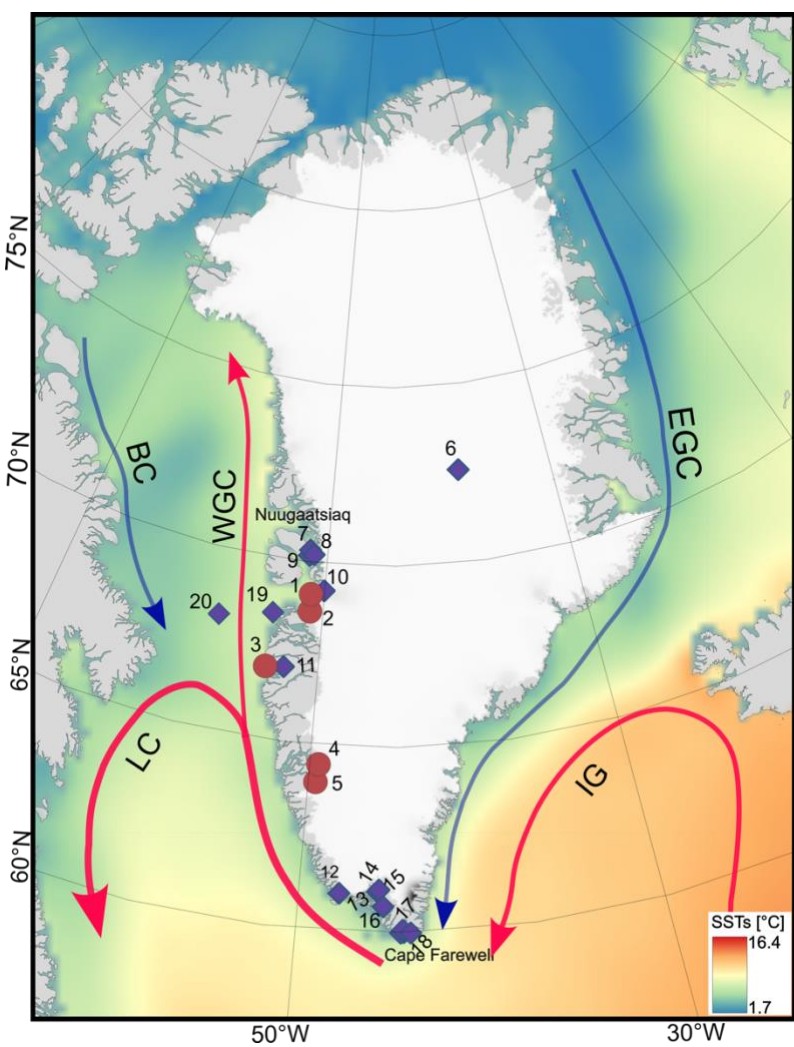

**Figure 1: Map of Greenland and surrounding area showing study lakes in brown circles (1–5), sites mentioned in the text in blue diamonds (6–20) and ocean currents; WGC: West Greenland, EGC: East Greenland, IR: Irminger, BC: Baffin, and LC: Labrador current. (1) Pluto, (2) N3, (3) Rosaea, (4) Marshall (5) Bullet (6) GISP2 (Kobashi et al., 2017), (7) Sermikassak, (8) Qangattaq ice cap (9) Saqqap Tasersua (Schweinsberg et al., 2017, 2019) (10) North Lake (Axford et al., 2013) (11) Lake Gus (Cluett et al., 2023) (12) Qipisarqo (Frechette and de Vernal, 2009; Kaplan et al., 2002) (13) Lake 578 (Schneider et al., 2024) (14) Spongilla Sø (Fredskild, 1973; Gajewski, 2015) (15) Alakariqssoq glaciers (Larocca et al., 2020) (16) Comarum Sø (Fredskild, 1973; Gajewski, 2015) (17) N14 (Andresen et al., 2004), (18) Quvnerit glaciers (Larocca et al., 2020), and marine sediment cores (19) MSM343300 (Ouellet-Bernier et al., 2014) and (20) CC70 (Gibb et al., 2015). The fill colour represents modern summer sea surface temperatures (SSTs), data source: (\*Fisher et al., 2023; Moon et al., 2023)**.



## 2.3 Biomarker analysis

We used the method described in Cluett et al. (2023) for the extraction and purification of GDGTs. Briefly, total lipids from homogenized 0.5 to 2 g of bulk sediment samples were extracted using a Dionex Accelerated Solvent Extractor 200 with 30 ml of dicholoromethane (DCM): methanol 9:1 (volume:volume, v:v). Afterwards, 0.1 ug of a $C_{46}$ glycerol dialkyl glycerol tetraether (GDGT) was added to the total lipid extracts (TLE). TLEs were separated into neutral and acid fractions using aminopropyl silica gel columns eluted with DCM:isopropanol (2:1 v/v) and acids with 4% acetic acid in DCM, respectively. Neutral fractions were further split into apolar, ketone and polar fractions using alumina columns eluted with hexane/DCM (9:1, v:v), hexane/DCM (1:1, v:v), and DCM/MeOH (1:1, v:v), respectively. GDGTs contained in the polar fraction were filtered using 0.45 µm PTFE filters. GDGTs were analyzed on an Agilent 1260 Ultra High-Performance Liquid Chromatograph (UHPLC, Agilent 1260) coupled to an Agilent 6120 Mass Selective Detector (MSD) in selected ion monitoring (SIM) mode in the Biogeochemistry Laboratory at the University of Massachusetts Amherst. For the separation of 5-methyl and 6-methyl isomers, we used two coupled waters UHPLC columns (150 mm * 2.1mm*1.7 µm) and followed the procedure described by Hopmans et al. (2016). Chromatography was performed applying a three-phase isocratic solvent gradient using 100% hexane (solvent A) and hexane/isopropanol (9:1, v:v, solvent B). Peak integration was done using the "chromatoPy" package in Python (Otiniano et al., in preparation).

We analyzed 23 samples from Pluto Lake at ~4 cm resolution, 67 samples from N3 Lake at ~3 cm resolution, 39 samples from Rosaea Lake at ~6 cm resolution, 23 samples from Marshall Lake at ~4 cm resolution, and 22 samples from Bullet Lake at ~7 cm resolution. All samples were from 0.5-cm-thick subsamples of sediment cores. The GDGT time series from N3 Lake is at 100-year-resolution for the past 8 ka. Those from lakes Pluto, Marshall, and Bullet provide 400-year-resolution, covering from 9 to 2 ka, 9 to 1 ka, and 9 to 1 ka, respectively. The GDGT time series from Lake Rosaea offers a 200-year-resolution, spanning from 9 to 1 ka.

## 2.4 Lake modelling

We ran sensitivity tests of the lake energy and water balance model developed by (Dee et al., 2018; Hostetler and Bartlein, 1990) and modified by (Morrill et al., 2019). We used the ERA5 climate data (air temperature, relative humidity, wind speed, incoming surface shortwave radiation, downward longwave radiation, surface pressure, and precipitation amount) averaged over all the grid points in the range 47.90°W to 53.90° E and 60.84°N to 70.84° N for January 1, 1994, to December 31, 2024, as meteorological input data (Muñoz Sabater, 2019). All the meteorological inputs were at a 6-hourly timestep except total precipitation amount, which is hourly timesteps summed to total 6-hourly precipitation amount. Lake surface areas were estimated using Google Earth, and catchment areas using Arctic DEM in QGIS (Porter et al., 2022). Depth-area slices at a resolution of 1 meter were estimated using bathymetry of each lake, which was determined either using sonar or by depth sounding with a weighted measuring tape. We ran the lake model for 30 years, with a 10-year spin-up period at the beginning using 1994 meteorological input data. Since the modern observations from the studied lakes are limited, we used the same




input parameters, including the shortwave coefficient, the fraction of advected air, and the neutral drag coefficient, for all lakes, as per a previously published model for Lake N3 (Corcoran et al., 2021). We acknowledge that using the same lake model parameters might lead to data-model mismatches with lake conditions, however for the purpose of sensitivity tests on
orbital scales, such changes can only have a minor impact. To evaluate the sensitivity of lakes to changes in air temperature we perturbed the ERA5 input annual, and summer (defined as June to September) temperature by ±10, ±6, ±4, and ±2 °C. To evaluate the impact of shading on the lake water temperature, we perturbed the incoming short-wave radiation by ±10, ±25, and +50%. We calculated shading for each lake using GDAL in python using Arctic DEM for daytime (sunset to sunrise) at 15-minute time steps every 10 days from April to September under modern insolation parameters. Wind direction at each lake
was calculated using 6-hourly ERA5 climate data from January 1, 1994, to December 31, 2024.

## 3. Results and discussion

### 3.1 Anoxia, sources of brGDGTs and temperature implications

Total brGDGT concentration in these lake sediments from southwestern Greenland is about 6 to 10 times higher than total isoprenoid GDGT (isoGDGT), with both revealing a similar downcore pattern in all lakes, peaking in the early Holocene and
declining towards the late Holocene (supplementary fig. 6–10). Caldarchaeol (GDGT-0) is the most abundant isoGDGT (supplementary fig. 11a), followed by crenarchaeol (GDGT-4) and GDGT-1, whereas GDGT-2, GDGT-3, and the crenarchaeol isomer (GDGT-4′) were below detection limit in many samples, restricting $TEX_{86}$-based temperature reconstruction (Schouten et al., 2002). Caldarchaeol is produced in anaerobic conditions in lake water or sediments by methanogenic *Euryarcheota* or heterotrophic *Bathyarchaea* (Buckles et al., 2013; Sinninghe Damsté et al., 2009), whereas
crenarchaeol is produced in aerobic lake water and sediments by *Thaumarchaeota* (Baxter et al., 2021; Besseling et al., 2018). Therefore the ratio of caldarchaeol to crenarchaeol (Cald/Cren) may be a sensitive indicator of suboxic conditions in the water column and/or lake sediments, where the isoGDGTs are produced (Blaga et al., 2009). Cald/Cren ranges from 0.8 to 595.7 in all study lakes, with generally low values in lakes Marshall and Rosaea and high values in lakes Bullet, N3 and Pluto (supplementary fig. 6–10). Cald/Cren was higher during the early Holocene compared to the late Holocene in lakes Pluto,
Rosaea, Bullet, and Marshall, which may indicate lower oxygen availability (Acharya et al., 2023; Baxter et al., 2021; Blaga et al., 2009) (supplementary fig. 6–10). After approx. 6 ka, the Cald/Cren remained relatively low and stable, indicating oxic lake sediments and/or water column. Cald/Cren in Lake N3 had higher values indicating suboxic conditions from 8 to 5 ka, and lags the decrease recorded in other lakes by about a millennium (supplementary fig. 6–10). We hypothesize that this could be related to a greater maximum depth (~17 m) of N3 compared to other lakes (<10 m water depth). Deeper conditions favor
persistent stratification, leading to prolonged suboxic conditions (Kerimoglu and Rinke, 2013). This early Holocene timing of high Cald/Cren is similar to results from other Arctic lakes (Cluett et al., 2023; McFarlin et al., 2023; Schneider et al., 2024) and is probably caused by prolonged thermal stratification due to greater ice cover duration and/or summer warming due to high summer insolation (Klanten et al., 2023; MacIntyre et al., 2018). Lake model simulations indicate that persistent summer





stratification is unlikely to occur in the study lakes under Holocene temperature conditions; rather they experience prolonged

isothermal mixing under annual or summer temperatures up to ~10 degrees warmer than modern (supplementary fig. 12). This

indicates that the sub-oxic conditions in the studied lakes likely occurred only during periods of ice cover, favoring the growth

of methanogenic *Euryarchaeota* or heterotrophic *Bathyarchaea* during winter (Baxter et al., 2021; Blaga et al., 2009; Daniels

et al., 2022). Winter sub-oxia would have a minimal impact on brGDGTs, which are primarily produced during the ice-free

period (Loomis et al., 2014b; Miller et al., 2018; Zhao et al., 2021). Therefore, we assume that brGDGT distributions are

minimally influenced by sub-oxic conditions.

BrGDGTs in lake sediments can be allochthonous – produced in the catchment soil and transported to the lake via runoff

erosion, and autochthonous – produced in the water column and surface sediment (Baxter et al., 2021; Tierney et al., 2012).

BrGDGTs produced in these diverse environments can have different responses to temperature, directly impacting the

temperature reconstruction (Acharya et al., 2023; Martin et al., 2020). Therefore, understanding the sources of brGDGTs is

crucial before temperature reconstruction. We evaluated the sources of brGDGTs in southwestern Greenland lakes by

comparing their distributions with those from high latitude (>50 °N) soil, peat, and lake sediment samples (Raberg et al., 2022).

BrGDGTs from southwestern Greenland lakes have different fractional abundance of hexa-, penta- and tetra-methylated

brGDGTs than those in peat and soil samples but are similar to high-latitude lake surface sediment samples (supplementary

fig. 13), suggesting lacustrine origin. Furthermore, brGDGT IIIa, which is produced in higher relative abundance in lake water

than in soils (Weber et al., 2018), is abundant in the southwestern Greenland lake sediments, similar to high-latitude lake

surface sediments, and greater than in high-latitude soils (supplementary fig. 11). This corroborates the autochthonous

production of brGDGTs in southwestern Greenland lakes, although there remains a possibility of minor brGDGT contributions

from the surrounding catchment.

While there are several lacustrine brGDGT–temperature calibrations available globally (Martínez-Sosa et al., 2021; Raberg et

al., 2021; Zhao et al., 2023), regionally (Bauersachs et al., 2023; Dang et al., 2018; Otiniano et al., 2023, 2024; Russell et al.,

2018), and site-specifically (Zhao et al., 2021; Bittner et al., 2022; Feng et al., 2019), only Raberg et al. (2021), Zhao et al.

(2021), and Otiniano et al. (2023; 2024) are based on the HPLC method separating GDGTs isomers and have considered the

Arctic seasonal climate. Therefore, we assess these calibrations to infer temperature in the southwestern Greenland lakes. The

Raberg et al. (2021) and Otiniano et al. (2023; 2024) calibrations estimate mean air temperature for the months above freezing

(MAF) while Zhao et al. (2021) estimates epilimnion lake water temperature for the ice-free season (Ice free season lake water

temperature, IFS LWT). For all analyzed samples, the reconstructed MAF based on Raberg et al. (2021) yields a temperature

range from ~3.9 to 10.9 °C; using Otiniano et al. (2024) yields a temperature range from 3.8 to 8.4 °C; and using Otiniano et

al. (2023) yields a temperature range from 5.8 to 10.0 °C, while IFS LWT reconstrued using Zhao et al. (2021) yields a

temperature range from ~6.0 to 21.8 °C (supplementary fig. 14). For lakes Marshall, N3, Pluto and Rosaea, all four calibrations

generated similar temperature trends throughout the Holocene, while the absolute values are different. For Bullet Lake, the



MBT'$_{5ME}$-based calibrations (Otiniano et al. (2024), Otiniano et al. (2023), and Zhao et al. (2021)) yielded similar trends, whereas Raberg et al. (2021) yielded a divergent trend, albeit with all the calibrations providing different absolute values. Since the calibration from Zhao et al. (2021) is specifically developed in a Greenland lake, and the brGDGTs in the five lakes studied here are lacustrine, we prefer to use this calibration to reconstruct IFS LWT in our study. This calibration has a root

mean square error of 0.5 °C. The IFS-LWT inferred for the core-top sample from Lake N3 is similar to the modern measurement of July Lake surface water temperature (supplementary fig. 14), implying the reliability of the Zhao et al. (2021) calibration for reconstructing IFS LWT in these lakes.

The isomer ratio of 5- to 6-methyl brGDGTs (IR$_{6Me}$) ranges from 0.13 to 0.51 across all study lakes (supplementary fig. 6–10). Previously, a non-thermal effect on lacustrine brGDGTs has been identified when IR$_{6Me}$ is greater than 0.4 (Bauersachs

et al., 2023; Novak et al., 2025). In southwestern Greenland, samples with IR$_{6Me}$ ≥ 0.3 correspond with abrupt (> ± 3 °C) temperature changes in single samples, suggesting these samples may be influenced by non-thermal effects (supplementary fig. 6–10 and supplementary fig. 14). Consequently, all samples with an IR$_{6ME}$ ≥ 0.3 were removed from temperature reconstructions from all study lakes (supplementary fig. 14).

**3.2 Holocene temperature variability**

Reconstructed IFS LWT show both similarities and differences between time series, with differences not following a latitudinal gradient. Reconstructed IFS LWT varies from 12.7 to 19.3 °C for Pluto, 9.5 to 18.2 °C for N3, 7.9 to 15.8 °C for Rosaea, 11.0 to 21.8 °C for Marshall, and 6.0 to 12.2 °C for Bullet, respectively (Fig. 2). In Pluto Lake, the temperature timeseries starts at approx. 7 ka, with warm conditions (Fig. 2a). In Marshall, Bullet, and Rosaea Lakes, the temperature timeseries begin from *ca.*9.5 ka, with warm conditions indicated in Marshall Lake, while cold conditions prevail in Bullet and Rosaea Lakes (Fig. 2b

and c). In Marshall Lake, temperature decreased around 8 ka and then increased, peaking at approx. 7 ka. In Lake N3, the temperature timeseries begins at 8 ka and reveals a trend towards warm conditions. In lakes N3 and Marshall, maximum temperature occurred between 7 to 5 ka, and in Pluto Lake peak warmth occurred from the beginning of the temperature timeseries at *ca.*6.5 ka to 5 ka (Fig. 2). Immediately following the peak warmth, temperatures from Lake Marshall reveal a gradual cooling trend towards present. In contrast, reconstructed temperatures from Lakes N3 and Pluto show a steady decrease

from approx. 5 to 4 ka, followed by warm conditions until 3 ka, then gradual cooling. In Bullet Lake, the temperature trends are muted compared to the other lakes. The temperature maximum began at the same time as these other sites, around 8 ka, but lasted longer, until 4 ka, when cooler conditions lasted until ~3 ka, followed by slight warming until ~1.5 ka and then cooling. Overall, the pattern of reconstructed IFS LWT from these four lakes, spanning 5° latitude in southwestern Greenland, is similar to the annual temperature reconstruction from Greenland Ice Sheet Project 2 ice core (Fig. 3b), albeit with differences

likely caused by resolution of the timeseries (Kobashi et al., 2017). Furthermore, the long-term temperature trends observed in N3, Pluto and Marshall lakes is comparable to brGDGT-inferred IFS LWT timeseries from Lake Gus (Cluett et al., 2023)





and Lake 578 (Schneider et al., 2024) and pollen-based summer temperature reconstruction from Qipisarqo Lake (Frechette and de Vernal, 2009) (Figs. 2, 3).

In contrast to the other timeseries, the reconstructed temperature in Lake Rosaea reveals consistent cool conditions from 9 to 275 7 ka, warming from 7 to 5 ka, and peak warmth between approx. 5 and 3 ka, lagging by two millennia peak warmth inferred from brGDGTs at six other southwestern Greenland lakes (Fig. 2b). This delayed peak warmth is synchronous with a biogenic silica record from Qipisarqo lake (Fig. 3i) (Kaplan et al., 2002) and pollen-based records from southern Greenland (not shown) (Frechette and de Vernal, 2009; Fredskild, 1973; Gajewski, 2015). Next, we discuss possible mechanisms explaining this temporal discrepancy in thermal maxima.

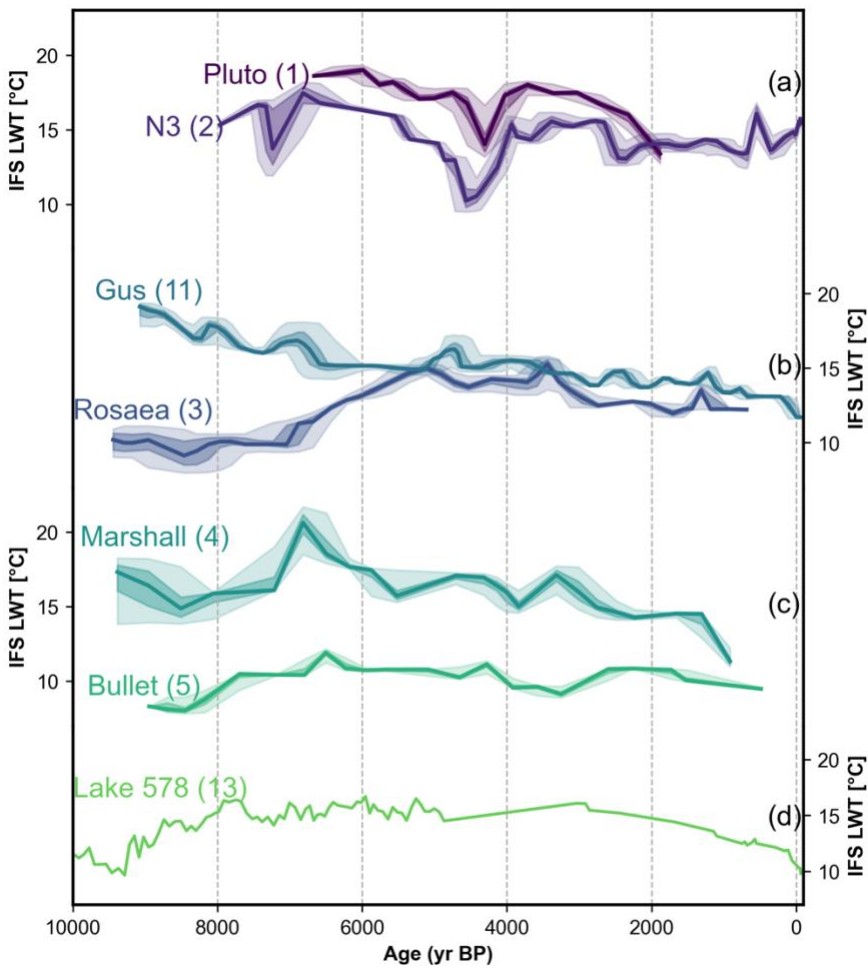


**Figure 2: Reconstructed ice free season lake water temperatures (IFS LWT) in (a) Lakes N3 and Pluto at 69 °N, (b) Lakes Gus (Cluett et al., 2023) and Rosaea at 67 °N, (c) Lakes Marshall and Bullet at 64 °N, and (d) Lake 578 at 61 °N (Schneider et al., 2024).**





**For timeseries in (a–c, bold line indicate median modeled age, and light and dark shading represetns 95[th] percentile and interquartile age uncertainty, respectively. Numbers in parentheses next to site name refer to site IDs shown in Fig. 1.**

### 3.3 Possible mechanisms causing temporal differences in peak warmth among time series on western Greenland

BrGDGT-inferred IFS LWT timeseries from southwestern Greenland reveal lower absolute value and magnitude of temperature change at Lake Bullet and delayed thermal maximum at Lake Rosaea, compared to the other lakes. To interpret these variations, we explored potential mechanisms influencing the brGDGT timeseries from the seven lakes in southwestern Greenland.

### 3.3.1 Production period of brGDGTs

Air temperature in southwestern Greenland is above freezing only from June to September, which causes lakes to become ice free and causes increases in lake water column nutrient content and light and oxygen availability (Shanahan et al., 2013; Zhao et al., 2021), in turn causing high aquatic production. Although the exact source organism(s) of lacustrine brGDGTs is still unclear, brGDGT flux in Arctic lakes is high during the ice-free season (Colcord et al., 2015; Zhao et al., 2021). Because lacustrine brGDGTs are primarily produced in the epilimnion during the ice-free season, differences between lakes, and between climate regimes, in the onset and the length of ice-free season can influence the timing of brGDGT production (Loomis et al., 2014a; Miller et al., 2018). Simulations of our five study lakes under 30 years of modern climate suggest that all lakes exhibit only minor (< 1 week) inter-lake variations in the timing of the onset and duration of the ice-free season (Supplementary fig. 15). Additionally, both the onset and duration of the ice-free season responds similarly in all five lakes to perturbations in annual air temperature (Figs. 4a– 4b). Annual, ice-free season and JJA surface water temperature of all the lakes vary similarly with changes in duration of the ice-free season (Figs. 4c – 4e). These results suggest that the temperature response of studied lakes to the same forcing is similar, and therefore if brGDGTs are produced during the ice-free season, their temperature response should be similar among lakes if atmospheric temperature change is similar.









**Figure 3: Holocene temperature history from southwestern Greenland. (a) Smaller-than-present mountain glaciers (Schweinsberg et al., 2017, 2019), (b) annual air temperature [°C] inferred from gas fractionation from Greenland Ice Sheet Project 2 ice core (Kobashi et al., 2017), (c) chironomid-inferred July air temperature [°C] from North Lake (Axford et al., 2013), brGDGT-inferred ice free season lake water temperature (IFS LWT) [°C] from (d) lakes N3 and Pluto (This study), (e) Lake Gus (Cluett et al., 2023), (f) lakes Marshall and Bullet (This study), brGDGTs-inferred ice free season lake water temperature (IFS-LWT) [°C] from (g) lake 578 (Schneider et al., 2024), (h) Lake Rosaea (This study), (i) Biogenic silica as an indicator of temperature from Lake Qipisarqo (Kaplan et al., 2002), Dinocyst-inferred sea surface temperatures (SST) [°C] from (j) Disko Bugt core MSM343300 (Ouellet-Bernier et al., 2014), and (k) Baffin Bay core CC70 (Gibb et al., 2015) (l) Greenland Ice Sheet surface area (\*10$^{12}$) [m$^2$] (Larsen et al., 2015) and (m) 21$^{st}$ June insolation at 65 °N [W m$^{-2}$] (Laskar et al., 2004). Numbers in parentheses next to site name refer to site IDs shown in Fig. 1.**

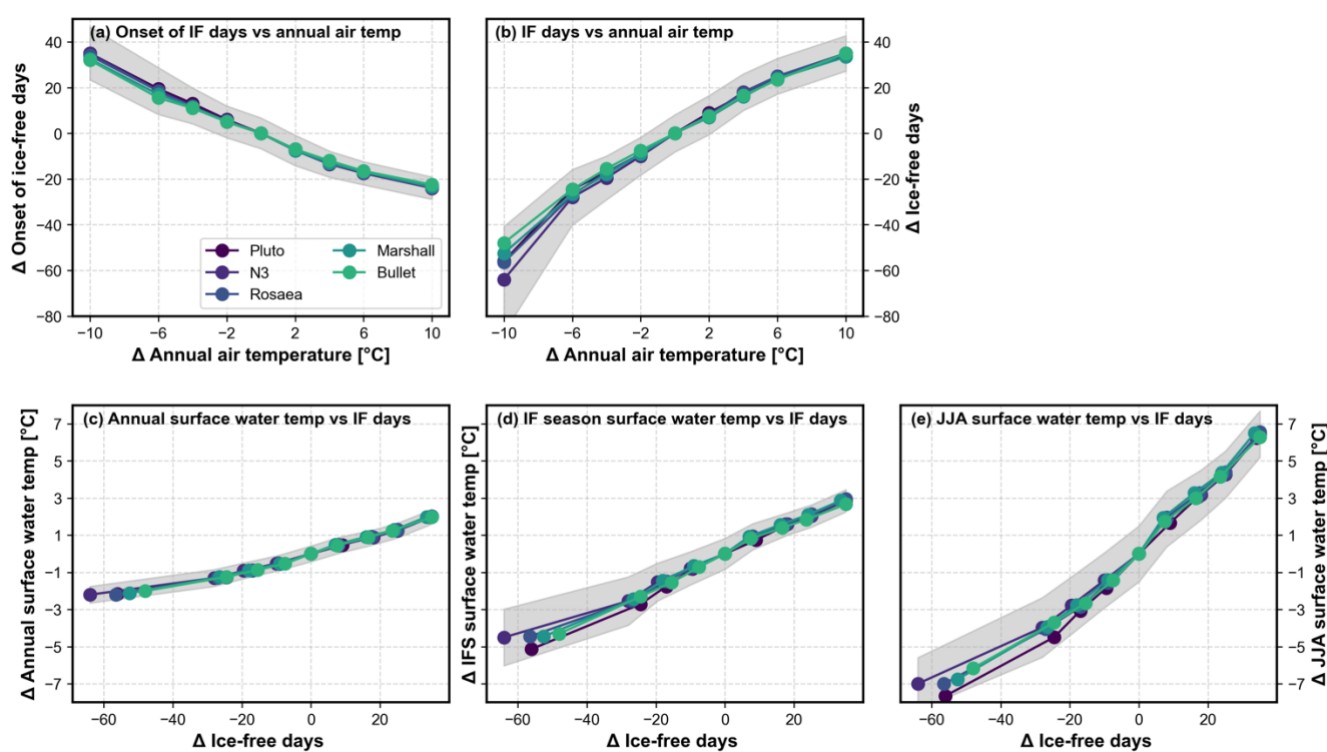

**Figure 4: Simulations of ice-free season and surface water temperature. (a) Sensitivity of changes in median onset date of ice-free conditions to annual air temperature [°C]. (b) Sensitivity of median ice-free duration to changes in annual air temperature [°C]. Changes in (c) annual, (d) ice free (IF), and (e) JJA surface water temperature [°C], with changes in duration of ice free conditions. Gray shading represents the standard deviation for the 30 years of model runs for Lake N3. Standard derivation is on the same order for other lakes.**



### 3.3.2 Sensitivity of lake system

Similar to the onset and duration of the ice-free season, the response of lake water temperature during the ice-free season may vary among study lakes due to their different morphological characteristics and thermodynamics. Lake model simulations under perturbed annual air temperatures reveal that the annual, ice-free days, and JJA surface water temperatures of the lakes 325 exhibit similar variations for the 30 years of model runs (Fig. 5). We obtained a similar, but slightly muted, response to June-September (JJAS or summer) air temperature perturbations (supplementary fig. 16). This result suggests that despite morphological differences the surface water temperature of the studied lakes, is equally sensitive to changes in air temperature. Since brGDGTs in lakes can be produced in surface waters, water columns, and/or sediment (Acharya et al., 2023; Buckles et al., 2014; Yao et al., 2020; Zhao et al., 2021), we also examined the response of the average depth profile temperature of the 330 lakes to perturbations in annual and JJAS air temperature (Fig. 5 d–f and supplementary fig. 16 d–f). As with the surface temperature, all the lakes exhibit similar responses, except Lake N3, which has a slightly muted response compared other lakes, especially in the JJA temperature profile, probably caused by greater maximum depth compared to other lakes. Therefore, the greater depth of Lake N3 may explain its slightly lower-amplitude (approx. <1.5 °C) Holocene temperature variations compared to the other lakes (Fig. 2). However, several millennia of delayed thermal maximum in Lake Rosaea and 335 approx. 4 °C lower absolute temperatures in Bullet Lake compared to other lakes cannot be explained by sensitivity to air temperature forcing.





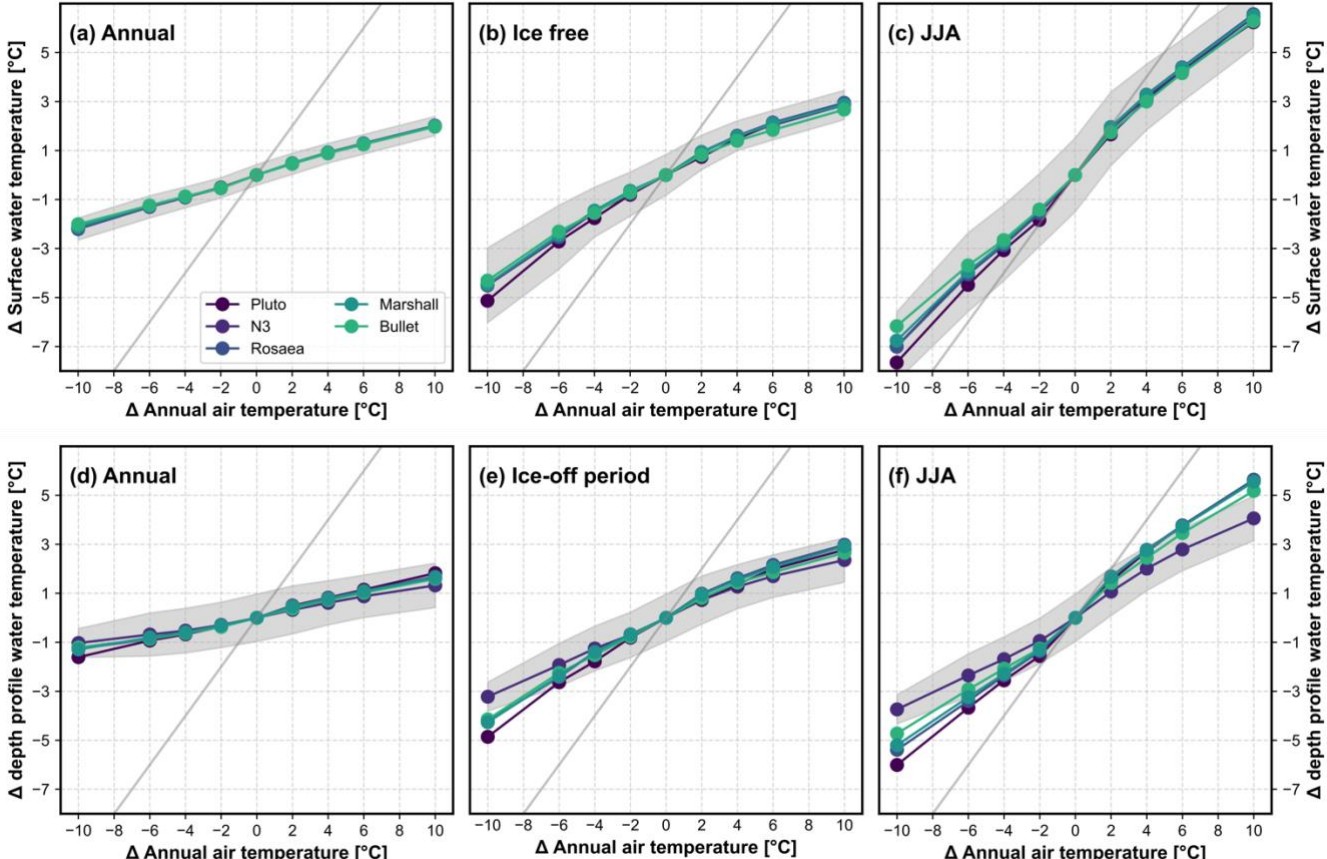

**Figure 5: Sensitivity of average surface water (a–c) and depth profile temperatures (d–f) to changes in annual air temperature.**
**Changes in (a) annual, (b) ice free, and (c) JJA surface water temperature [°C], with changes in annual air temperature. Changes**
**in (D) annual, (E) ice free, and (F) JJA average depth profile water temperature [°C], with changes in annual air temperature. Gray**
**line indicates the 1:1 line. Gray shading represents the standard deviation of 30 years of model runs for Lake N3. Standard derivation**
**is on the same order for other lakes.**

### 3.3.3 North-South pattern in timing of temperature maxima

To explore the north-south pattern in the timing of temperature maxima, we compared all the available brGDGT-based
temperature reconstructions from southwestern Greenland including from Lake Gus (Fig. 1; point 11) (Cluett et al., 2023) and
Lake 578 (Fig. 1; point 13) (Schneider et al., 2024), as these sites likely reflect similar seasonality as the lakes in this study.
These seven lakes cover the latitudinal gradient from ~61°N to 69°N (Table 1). If we assume a north-south pattern in the timing
and magnitude of the temperature maxima caused by Arctic Amplification (Axford et al., 2021; Briner et al., 2016; Kaufman,
2004), the lakes at the highest latitudes in the transect (Pluto and N3) should exhibit higher magnitude and earlier temperature
maxima compared to lakes at the lower latitudes (Rosaea and Gus, 67°N; Marshall and Bullet 64°N; and 578, 61°N). However,



the timing of the temperature maximum at the southernmost site, Lake 578 (8 to 5 ka) is similar to that of lakes N3 and Pluto at 69 °N and Lake Marshall at 64 °N, occurring between *ca.* 7 to 5 ka (Fig. 2). Moreover, Lake Bullet, situated at a similar
latitude to that of Lake Marshall, exhibits a prolonged temperature maximum between *ca.* 8 to 4 ka. The time series that shows a delayed temperature maximum, between *ca.* 5 and 3 ka, occurs in the middle of the transect: at Lake Rosaea (67°N). This delayed timing in temperature maximum at Lake Rosaea is not observed in the timeseries from Lake Gus, at the same latitude, where peak temperatures occurred between *ca.* 9 to 6 ka (Cluett et al., 2023). A similar thermal maximum timing among six of the seven brGDGT timeseries from southwestern Greenland suggests that Arctic Amplification was not the primary
mechanism setting the timing of peak warmth in this region. In addition, Lake Rosaea must be primarily responding to different primary atmospheric forcing mechanisms than the other lakes.

### 3.3.4 Ice-sheet and ocean proximity

Ice sheets have high albedo and can generate cold katabatic winds draining downslope, leading to a locally colder climate (van den Broeke et al., 1994; Laffin et al., 2023). Lakes Bullet and Marshall are <2 km from the modern Greenland Ice Sheet,
whereas lakes N3, 578, and Pluto are > 10 km, and lakes Gus and Rosaea are > 100 km away (Table 1). Deglaciation from lakes Gus, Rosaea, Marshall, Bullet, and 578 occurred before 10 ka (Leger et al., 2024). The Greenland Ice Sheet was relatively stable from 9 to 6 ka near lakes Marshall, Bullet, and 578, while it was receding inland from lakes Rosaea and Gus. Since 9 ka, the ice sheet was always at least 50 km inland from lakes Rosaea and Gus. By contrast, lakes N3 and Pluto deglaciated at ~8 ka (Leger et al., 2024; Young et al., 2013) and the Greenland Ice Sheet was <3 km aerial distance from these lakes until 7.5
ka. After 6 ka, the southwestern Greenland Ice Sheet retreated inland of its present-day margin and returned to its 6 ka margin around 1850 AD (Leger et al., 2024), although the little available evidence suggests the retreat was <10 km (Briner et al., 2014; Cronauer et al., 2016; Lesnek et al., 2020). This suggests that lakes Bullet, Marshall, Pluto, and N3 could have been significantly influenced by the proximity of the ice sheet during the Holocene, while Lakes Rosaea and Gus are minimally influenced, and Lake 578 might be moderately influenced. Thus, the fact that peak warmth occurred after 8 ka, following ice
sheet retreat from Lakes Marshall, Bullet, N3, and Pluto suggests that the ice sheet may have kept local conditions cold. However, it should be noted that Marshall Lake experienced moderately warm conditions at 9 ka, a period that most of these other timeseries do not capture. Lake Rosaea is an obvious outlier in this ice-sheet proximity pattern: it was far from the ice sheet for most of the Holocene, so should be minimally influenced by katabatic winds, but has a low magnitude of temperature change and the latest thermal maximum of all the records. Thus, temperature at Lake Rosaea is likely influenced by additional
factors.

Ocean conditions influence coastal climate by affecting ocean-atmosphere heat transfer (Bashmachnikov et al., 2023). A warmer ocean releases heat into the atmosphere, resulting in warmer coastal climates, and vice versa with cold ocean conditions. Lakes Bullet and Marshall are ≥100 km and lakes 578 and Gus are >60 km from the open ocean (Table 1). Thus, the ocean influence on these lakes is likely minimal. Lakes N3, Pluto, and Rosaea are situated near the ocean (<3 km). Modern



wind directions at Lake Rosaea are predominantly southerly, meaning that most airmasses at Lake Rosea come directly from the Labrador Sea and Davis Strait. About 20% of winds at Lake Rosea are northerly, meaning they pass over Baffin Bay prior to arriving at Lake Rosea. In contrast, at lakes Pluto and N3, wind direction is more variable, meaning that half of the airmasses originate from land and half from Disko Bugt (Fig. 6a). The pattern of reconstructed lake water temperature from Lake Rosaea is similar to summer sea surface temperature (SST) timeseries from Disko Bugt and Baffin Bay (Fig. 5j, k), as well as other

North Atlantic SST time series, which reach maxima in the middle and late Holocene (Gibb et al., 2015; Hansen et al., 2020; Ouellet-Bernier et al., 2014; Saini et al., 2022). This similarity implies that temperature at Lake Rosaea is more strongly influenced by oceanic conditions via ocean-atmosphere heat exchange than the other lakes in this transect. Summer SSTs in eastern Baffin Bay were cold during the early Holocene, probably caused by meltwater from the rapidly retreating Greenland Ice Sheet and a weaker western Greenland current (Gibb et al., 2015; Hansen et al., 2020). SSTs rose during the middle-

Holocene related to decreased meltwater from the Greenland Ice Sheet and increased influence of warm Atlantic water masses (Ouellet-Bernier et al., 2014; Perner et al., 2013). The reconstructed temperature pattern from Lakes Pluto and N3 exhibits a closer resemblance to the lakes that are far from the ocean. These lakes may have been less affected by Holocene Ocean conditions than Lake Rosaea due to local topography, including proximity to the ice sheet and their position in interior Disko Bugt.




### 3.3.5 Topographic conditions

If there is steep topography surrounding a lake, this can shade the lake, influencing the direct short-wave radiation reaching lake surface, thereby regulating the lake surface water temperatures. To investigate the mechanisms for the differences in absolute temperature and magnitude of Holocene temperature variations in the five study lakes, we examined the topographic

conditions around each lake. Shading calculations show that of the five lakes, Pluto Lake is the least shaded by surrounding topography and Bullet Lake is the most shaded (15% less direct shortwave radiation than Pluto Lake) (Fig. 6b). Lakes N3 and Marshall are approx. 5% more shaded than Pluto Lake, whereas Lake Rosaea is 10% more shaded than Pluto Lake (Fig. 6b). This means that the topography surrounding lakes N3, Marshall, Rosaea, and Bullet cause them to receive correspondingly less solar radiation compared to Pluto, with Bullet receiving the least. We assess the impact of shading on lake water

thermodynamics by perturbing the magnitude of incoming shortwave radiation in the lake model. These simulations suggest that a 10% decrease in incoming short-wave radiation can suppress the mean ice-free season lake surface temperature by approx. 1.5 °C and a 25% decrease by 4 °C (Fig. 6c). Therefore, we suggest that the slight (~1–2 °C) difference in absolute temperature values between the lakes Pluto, N3, Marshall and Rosaea could be related to local topography and variations in incoming shortwave radiation to the lakes. Approx. 4 °C lower absolute temperature and approx. 1 °C lower magnitude of

temperature change in Bullet may be due to lowest incoming shortwave radiation to the lake, compared to the other study lakes. There may also be other factors causing differences between these time series, but we believe we have assessed the primary mechanisms here.





Figure 6: (a) Wind strength and direction in the study lakes from April to September, based on six-hourly ERA5 data from 1994 to 2024. (b) Percentage change in average summer (April to September) direct radiation on study lakes relative to Pluto Lake, which is the least shaded of the five. (c) Sensitivity of average surface water temperature to changes in incoming shortwave radiation. Gray shading represents the standard deviation of 30 years of model runs for Lake N3. Standard derivation is on the same order for other lakes.



## 4 Holocene temperature history of southwestern Greenland

The majority of brGDGT-inferred temperature timeseries from southwestern Greenland (N3, Pluto, Marshall, Bullet and 578) document peak warmth in the middle Holocene, between *ca.* 7 and 5 ka, following the maximum solar insolation. Timeseries from Lake Gus reveal peak warmth between *ca.* 9 to 6 ka, earlier than many other time series. An earlier deglaciation at Lake Gus (before ca. 10 ka), and minimal influence from Greenland Ice Sheet and coast during Holocene, likely have caused earlier warmth. However, it should be noted that ~25 samples from early Holocene section from Gus Lake were removed due to evidence of high input of soil-derived brGDGTs (Cluett et al., 2023), so it is possible that other samples in that section have some minor influence of soil-derived brGDGTs. The timeseries from Lake Marshall shows warmth at *ca.* 9 ka, followed by cooling *ca.* 8 ka and subsequent warming, similar to the GISP2 record (Kobashi et al., 2017) and ice-sheet retreat rate reconstructions (Lesnek et al., 2020), suggesting early Holocene warm conditions not captured in most lake records. Immediately after peak warmth, temperature reconstructions show a gradual cooling trend towards the late Holocene. This pattern of temperature change can also be inferred from alpine glacier reconstructions from throughout southwestern Greenland, which experienced high retreat rates or completely disappeared during early Holocene and reappeared during late Holocene (Fig. 3a) (Larocca et al., 2020; Larsen et al., 2017; Lasher et al., 2020; Schweinsberg et al., 2017, 2019). The early Holocene peak warmth and the subsequent cooler late Holocene documented in southwestern Greenland align with the temperature pattern observed in many proxy time series from Greenland and Arctic Canada, albeit with spatial and temporal variability (Axford et al., 2021; Briner et al., 2016; Kaufman, 2004; Larocca and Axford, 2022).

Lake Rosaea, which is influenced by ocean-atmosphere heat exchange, experienced a thermal maximum between approx. 5 to 3 ka, coinciding with nearby SST reconstructions (Fig. 3j, k) (Gibb et al., 2015; Ouellet-Bernier et al., 2014). Similarly, the biogenic silica (BSi) record from Lake Qipisarqo (Fig. 3i), pollen-based reconstructions from Lakes Camarun Cø, and Kloft Sø, all coastal sites, reveal a delayed thermal maximum, after 6 ka, in accord with marine temperature timeseries (Fig. 3j, k) (Andresen et al., 2004; Fredskild, 1973; Gajewski, 2015; Kaplan et al., 2002). This suggests the influence of ocean conditions on coastal temperatures via ocean-atmosphere heat exchange. Despite the thermal maximum after 6 ka indicated by BSi from Qipisarqo lake (Kaplan et al., 2002), pollen-based reconstruction from the lake reveals an earlier thermal maximum in the past 9 ka between 7 to 5 ka (Frechette and de Vernal, 2009). As pollen can have a larger transport distance and provide a more regional signal, the discrepancy between the pollen and BSi could be related to pollen climate of regions upwind, rather than the local signal recorded by BSi or BSi timeseries is affected by dissolution. It should also be noted that caution is advised in interpretating pollen-based temperature reconstructions due to the potential impact of vegetation migration and/or no-analog climate (Axford et al., 2021).

The magnitude of peak warmth in brGDGT-inferred IFS LWT show both similarities and differences between time series. Excluding a point at 6.9 ka from Marshall Lake that shows approx. 4 °C warmer climate conditions compared to middle-Holocene samples and a point at 4.6 ka from N3 Lake that shows approx. 4 °C colder climate conditions compared to the





middle Holocene samples, brGDGT-inferred IFS LWT from Pluto, N3, Gus, Marshall, and Bullet Lake show average 1.4 ± 0.3 °C warmer temperatures during the peak warmth (from 7 to 5 ka for Pluto, N3, Marshall, and Bullet Lake, and from 9 to 6 ka for Gus Lake) relative to average temperature between 2 to 4 ka (Fig. 3). Bullet Lake show approx. 0.4 °C less warming

compared to the other lakes. The relatively colder temperature in Bullet Lake is likely due to reduced solar radiation received by the lake resulting from the surrounding topography. BrGDGT-inferred IFS LWT timeseries from Lake 578 is 1.0 °C warmer from 7 to 5 ka relative to average temperature between 2 to 4 ka. This temperate change is smaller than the observed change in the other lakes and may simply be due to low temporal resolution of the time series between 2 to 4 ka, anomaly period. Overall, the magnitude of peak warmth in lakes Pluto, N3, Gus, Marshall, and Bullet are comparable to the chironomid-inferred

July air temperature from North Lake showing 1 °C warmer conditions between 7 and 5 ka relative to 2 to 4 ka (Axford et al., 2013). The magnitude of peak warmth in timeseries from Pluto, N3, Gus, Marshall, and Bullet Lake is higher than the annual air temperature [°C] inferred from gas fractionation from GISP2 ice core demonstrating 0.15 °C warmer conditions from 7 to 5 ka relative to 2 to 4 ka (Kobashi et al., 2017). The higher magnitude of temperature change in timeseries from Pluto, N3, Gus, Marshall, and Bullet Lake compared to GISP2 could be due to the differences in temporal resolution of the timeseries,

the seasonality of reconstructed temperatures, and the influence of distinct atmospheric-oceanic and ice sheet feedback mechanisms between timeseries.

BrGDGT-inferred IFS LWT from Lake Rosaea do not show elevated warming during the peak warmth in the timeseries from 5 to 3 ka compared to 2 to 4 ka. This value is approx. 1°C lower than the peak warmth in the other study lakes. As Lake Rosaea is highly influenced by the oceanic conditions, the lower magnitude of temperature change in Rosaea Lake is due to lower

magnitude of temperature change in marine timeseries compared to terrestrial timeseries (Gibb et al., 2015; Hansen et al., 2020; Ouellet-Bernier et al., 2014).

Although there are differences in timing and magnitude of peak warmth in brGDGT-inferred from IFS LWT timeseries, they do not follow a latitudinal pattern. This is surprising, considering the pronounced warming in higher latitudes compared to low latitudes due to local feedback and remote teleconnections observed during anthropogenic warming period (England et al.,

2021; Sweeney et al., 2023) and during the Holocene (Axford et al., 2021; Briner et al., 2016; Kaufman, 2004). The absence of Arctic Amplification in brGDGT-inferred from IFS LWT timeseries in the past 9 ka could be because of several factors: (1) the latitudinal gradient across the study sites may be too small to observe temperature differences; (2) the temporal resolution of the time series may be too coarse to detect subtle shifts in the timing of the thermal maximum; or (3) other local or regional influences, including strong ocean and atmosphere northward heat transport (Dufour et al., 2016) and the absence of Holocene

summer sea ice in this region (Gibb et al., 2015), may have been stronger than the expected signal from Arctic amplification. Future high-resolution proxy timeseries covering a larger latitudinal range and using the same proxy, reflecting similar seasonality, could help test this hypothesis.




## 5 Summary and conclusions

We compare five new (from Pluto, N3, Rosaea, Marshall, and Bullet Lake) and two published (from Gus Lake and Lake 578)
brGDGT-inferred IFS LWT time series spanning the past 9 ka from southwestern Greenland. The timeseries from lakes Pluto, N3, Marshall, Bullet, and 578 have similar trends, with temperature maxima between *ca.* 7 and 5 ka, $1.4 \pm 0.3$ °C warmer than 2 to 4 ka, and then gradual cooling towards the late Holocene, similar to the many other regional timeseries, following maximum annual insolation at 65 °N. The timeseries from Lake Gus reveals peak warmth between *ca.* 9 to 6 ka, earlier than many other time series, likely due to minimal influence from the coast and the Greenland Ice Sheet. The time series from Lake
Marshall suggests a warm early Holocene, cooling around 8 ka and subsequent warming, which is indicated in the GISP2 temperature timeseries and reconstructions of ice-sheet retreat rates.

In contrast to other timeseries, Lake Rosaea reveals a temperature maximum between ca. 5 and 3 ka, lagging peak warmth in other timeseries by two millennia. Lake model simulations suggest that proxy systematics, i.e., the production season of brGDGTs and the sensitivity of each lake system to climate changes, are not the important factors causing temporal differences
between the time series. Moreover, we do not observe a north-south trend in timing of the thermal maximum in southwestern Greenland. Instead, the timing of peak warmth seems to be related to regional-scale factors: proximity to the ice sheet and ocean. Lake Rosaea is situated at the coast, with prevailing southerly winds from the ocean. Holocene temperature change at Lake Rosaea is synchronous with regional marine timeseries that indicate weaker West Greenland Current and cooler summer ocean conditions during the early and middle Holocene caused by melt water from Greenland Ice Sheet (Axford et al., 2021;
Gibb et al., 2015; Hansen et al., 2020; Ouellet-Bernier et al., 2014). Temperature at Lake Rosaea rose during the middle-Holocene when SSTs increased, due to decreased melt water flow from the Greenland Ice Sheet, which reached its minimum extent around 5 to 3 ka (Briner et al., 2014) and increased influence of warm Atlantic water masses (Gibb et al., 2015; Hansen et al., 2020; Ouellet-Bernier et al., 2014). Similar to Lake Rosaea, proxy timeseries from other coastal sites, including Lake Qipisarqo, Camarun Cø, and Kloft Sø, show a delayed thermal maximum, in accord with SST timeseries. We therefore suggest
that temperature reconstructions from coastal sites where prevailing winds are onshore may be strongly influenced by ocean conditions via ocean-atmosphere heat exchange.

Overall, brGDGT-inferred IFS LWT timeseries suggest that local variations in temperature, influenced by the proximity to the ice sheet and ocean, caused discrepancies in timing and magnitude of the Holocene temperature reconstructions in southwestern Greenland. Seasonality (e.g., IFS, summer, winter, annual) of reconstructed temperature seems important when
comparing between timeseries reconstructed using different proxies. Therefore, further investigation of proxy time series to quantify seasonal variability and the impact of local factors, e.g., site-specific, topographic, ice sheet proximity, and oceanic conditions, will likely reveal broad similarities among proxy timeseries, helping to reconcile differences and refine Holocene climate reconstructions.



**Data availability**

All the data will be available at National Centers for Environmental Information for Paleoclimatology data portal.

**Authors contribution**

The study was conceptualized by SA and EKT. AC, ALG, and ISC conducted the laboratory analysis. SA and EKT prepared the manuscript with contributions from all co-authors.

**Competing interests**

The authors declare that they have no conflict of interest

**Acknowledgements**

We thank John Ryan-Henry, Kayla Hollister, Owen Cowling, Nancy Leon, and Jeff Salacup for their help in lipid biomarker analysis. We gratefully acknowledge Dr. Gerard A. Otiniano and Dr. Samuel Mark for discussions.

**Financial support**

This research is funded by National Science Foundation grant ARCSS-2106971 to EKT and JPB.

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
