# Peer review of "Holocene temperatures in southwestern Greenland controlled by topography, ice sheet proximity and oceanic conditions"

_EGUsphere, 2025_

## Author Comment (AC1)

**Response to reviewer#1**

We would like to thank the Anonymous Referee#1 for insightful comments that will significantly improve our manuscript. Below we have addressed each comment one by one, with the original comments in back text and our response in blue text.

The use of brGDGT distributions to estimate temperature is extensively calibrated and commonly used, but the proxy is also widely documented to respond to other environmental gradients beyond temperature. In particular, there is a growing body of work showing that local lake conditions including redox status can overwhelm or skew temperature (Raberg et al., 2025; Yao et al., 2020; Zander et al., 2024) and evidence that Holocene brGDGT temperature reconstructions at Arctic sites are impacted (de Wet et al., 2019; Kusch et al., 2019; Lattaud et al., 2021). The authors acknowledge this as a potential confounding factor and their data show periods of elevated concentrations of isoGDGT0 along with extremely high isoGDGT0/Cren ("Cald/Cren") ratios (>200) that indicate methanogen production and thus anoxia is indeed heightened at some sites through parts of the Holocene (e.g., Figs. S6-S10). However, they quickly rule out the likelihood of brGDGT distributions being impacted by site-specific redox conditions, reflected in isoGDGTs, based on 1) A lake modeling exercise that indicates all sites are invulnerable to summer stratification, and thus 2) that isoGDGTs have a different production seasonality (winter, with anoxia driven by prolonged ice cover) than brGDGTs (summer). However, there are obvious problems with the lake model, and the second point lacks any additional data to support it independently, as described below.

**Lake Model**

The presented lake model output for the suite of lakes demonstrates every lake in the dataset is invulnerable to changes in summer mixing regime, even at summer air temperature perturbations as high as +10 degrees. Supplemental Fig. S16 indicates that +10 deg of JJA air temperature will yield just +2 degrees of IFS surface LWT change at every lake despite morphological and catchment type differences. This same model also shows invariant ice phenology regimes (Fig. S15) between the lakes, despite large differences in both elevation and latitude. These model outputs are contradicted by observational data at sites across Greenland and the Arctic broadly, which demonstrate A) lake water and air temperature are much more closely matched at most sites (Carrea et al., 2025; Kettle et al., 2004; Piccolroaz et al., 2020; Tong et al., 2023),

Reply: We appreciate the feedback on this point. On an intra-seasonal scale, lake surface temperature does change in step with air temperature, e.g. Fig 4. In Kettle et al., (2004), but the magnitude of lake temperature response is smaller and delayed compared to air temperature. This dampened response of lake surface temperature to air temperature is also evident on interannual time scales (e.g., Fig 4c in Piccolroaz et al 2020). Lakes at high latitudes have particularly low responses, changing by 0.2°C for every 1°C air temperature change (Piccolroaz et al., 2020). The lake model sensitivity tests that we conduct here suggest a lake temperature response of ~0.4 to 0.6°C for every 1°C annual

or JJAS air temperature change, in line with observations of global high-latitude lakes (Cluett et al., 2023; Piccolroaz et al., 2020). Thus, the model results are supported by temperature observations and therefore seem robust. We will clarify this point in the text as follows, in section 3.1:

"Lake model simulations under perturbed annual or JJAS air temperatures suggest a IFS LWT response of ~0.4 to 0.6°C for every 1°C annual or JJAS air temperature change, in line with observations of global high-latitude lakes (Cluett et al., 2023; Piccolroaz et al., 2020)."

B) Arctic lakes can be vulnerable to summer stratification at even modern warming levels (Antoniades et al., 2024) and with a temperature difference between the epi and hypolimnion as little as 0.5 deg C (Klanten et al., 2024), and there are comparable lakes in this sector of Greenland that are summer stratified today (Saros et al., 2016),

Reply: We agree with the reviewer that summer stratification occurs in Arctic lakes.

We will clarify this in the updated version of the manuscript as:

"Lake model simulations under 30 years of modern climate suggest the occurrence of stratification (i.e., mixing depth is less than the maximum lake depth) in all the study lakes during winter and early in the ice-free season (Fig. S16/Fig. 1 below), as observed today in Arctic lakes (Antoniades et al., 2024; Klanten et al., 2024; Saros et al., 2016). The duration of ice-free season stratification increases in all the study lakes with an increase in annual or JJAS temperature and vice versa for a temperature decrease (Fig. S16/Fig.2 below). However, the simulations show that, no matter the climate forcing, the lakes do not remain stratified throughout the entire ice-free season. This means that, even if the study lakes are stratified during early spring, the lake water fully mixes during the late summer and autumn, a phenomenon observed in Arctic lakes with similar size and chemical composition (Antoniades et al., 2024; Klanten et al., 2023, 2024; Lindborg et al., 2016). The duration of the isothermal mixing period changes similarly in all the study lakes in response to air temperature model sensitivity test, except for Lake N3, to perturbations in annual or JJAS air temperature (Fig. S16). In Lake N3, the duration of the isothermal mixing period shows minor variations with perturbation in annual or JJAS air temperate, probably because of its larger surface area and depth compared to other study lakes. Overall, these model simulations suggests that while summer stratification does occur in these lakes, the duration of stratification is relatively short and therefore likely unable to support summer suboxic conditions, similar to Arctic lakes that experience summer stratification today. The lakes in this study are relatively shallow and have low dissolved solids compared to some Arctic lakes that today experience extended summer stratification and hypolimetic sub-oxia (Antoniades et al., 2024; Klanten et al., 2023; Saros et al., 2016), making the study lakes in this study more susceptible to mixing by solar heating and wind. Winter sub-oxia would have a minimal impact on brGDGTs, which are primarily produced during the ice-free period (Loomis et al., 2014b; Shanahan et al., 2013; Zhao et al., 2021). Therefore, we assume that brGDGTs in the study lakes, mainly produced during summer, are minimally influenced by changes in sub-oxic conditions during the Holocene."

Figure 1: Simulated mixing depth in study lakes with changes in (A) annual air temperature, and (B) JJAS air temperature. Modern represents the model run under the current ERA5 climate conditions. All the model runs are plotted as the median of the 30-year periods. The initial spring increase in mixing depth coincides with onset of ice-free season in all simulations, and the fall decrease in mixing depth to 0 m coincides with the end of the ice-free season. Ice cover period stratification occurs when the maximum depth is less or equal to 1 meter. Ice-free season stratification refers to the period when the mixing depth is less than 1 until the 75% of the lake's maximum depth. Ice-free isothermal mixing period refers to the period when the mixing depth is greater than 75% of the lake's maximum depth.

**Figure 2:** Simulated duration of ice-free season stratification [days] in study lakes with changes in (A) annual air temperature [°C], and (B) JJAS air temperature [°C]. Modern represents the model run under the current ERA5 climate conditions. All the model runs are plotted as the median of the 30-year periods.

and C) ice phenology (i.e., timing and length of the ice-free season) is almost certainly variant at these sites based on the elevation range (Posch et al., 2024).

Reply: We agree with the reviewer that modern lake ice phenology is different between the study lakes. We updated our modeling scheme to use climate specific to each lake location (before we used the same average value for all lakes). We find that the updated climate data results in more reasonable lake ice phenology (Figure 3). Even so, all the lakes except N3 show a similar response to air temperature changes, suggesting that the morphometry of the lakes is more important than the climate differences in governing their response, at least across these spatial scales.

We will clarify this point in section 2.4 as follows:

"We used the ERA5 climate data (air temperature, relative humidity, wind speed, incoming surface shortwave radiation, downward longwave radiation, surface pressure, and precipitation amount) averaged over four grid boxes nearest of each lake for January 1, 1994, to December 31, 2024, as meteorological input data (Muñoz Sabater, 2019)."

We will clarify this point in the section 3.3.1 as follow:

"Simulations of our five study lakes under 30 years of modern climate suggest that all lakes exhibit

**Figure 3:** Median of simulated fraction of ice cover in study lakes for 30 years of model runs under modern conditions.

Additionally, this model indicates that Holocene-scale air temperature changes across Greenland (~+0-4 deg C) should result in changes that are barely detectable in brGDGT distributions (IFS LWT

**Figure 4:** A. Ratio of caldarchaeol to crenarchaeol, B. ratio of 5-methyl hexamethylated and pentamethylated brGDGTs with no cyclopentane rings (HP5), and fractional abundance of brGDGT Illa (%Illa) in southwestern Greenland lakes.

Table 1: Correlation (r-value and p-value) between the ratio of caldarchaeol to crenarchaeol (cald/cren) and three brGDGT indices: 1. fractional abundance of brGDGT IIIa (%IIIa), 2. ratio of 5-methyl hexamethylated and pentamethylated brGDGTs with no cyclopentane rings (HP5) index and 3. methylation index of branched tetraethers (MBT'5Me). Significant correlations (p < 0.01) are bolded.

| Lake     | cald/cren vs. %IIIa | cald/cren vs. HP5 | cald/cren vs. MBT' 5Me |
|----------|---------------------|-------------------|-----------------------------------|
| Bullet   | -0.30 (p=0.18)      | 0.16 (p=0.472)    | 0.08 (p=0.713)                    |
| Marshall | 0.31 (p=0.16)       | 0.43 (p=0.052)    | -0.40 (p=0.071)                   |
| Pluto    | 0.19 (p=0.38)       | 0.64 (p

**Figure 5:** Fractional abundance of brGDGTs IIIa" (Weber et al., 2015) in Lake Bullet sediment samples.

**We will revise the text as follow:**

"BrGDGT production in Arctic lakes is higher during the ice-free season than the ice covered season (Shanahan et al., 2013; Zhao et al., 2021). The brGDGT indices that can indicate production under suboxic conditions, HP5 and fractional abundance of brGDGT Illa (%Illa) (Weber et al., 2018; Yao et al., 2020), were high in the study lakes from 10 to 8 ka, low between 8 and 5 ka, and gradually increasing thereafter (Fig. 2b and Fig. S6–

10). In lakes Bullet, Marshall, N3, and Pluto, correlations between cald/cren and both HP5 and %Illa were not significant (Table S4), suggesting a decoupling between sub-oxic conditions inferred from isoGDGT-derived cald/cren and brGDGT-derived HP5 and %Illa. In Lake N3, we observed a significant negative correlation between GDGT-0/cren and %Illa, indicating antiphase sub-oxic conditions inferred from isoGDGT and brGDGT proxies (supplementary fig. S6-10 and supplementary table S4). In contrast, Lake Rosaea had significant positive correlations between cald/cren and both HP5 and %Illa. However, the cald/cren ratio in Lake Rosaea remained below 45 in all samples except the last sample where ratio is <110, suggesting relatively oxic conditions, although it is possible that suboxia infleuened the brGDGTs distributions in Lake Rosaea (Fig. S8). Based on the evidence for seasonal production and the contrasting trends in these lake time series, we interpret the br-and iso-GDGTs to be predominantly produced during different seasons: brGDGTs in ice-free season and isoGDGTs in ice covered season. This contrast in seasonal production may be enhanced when caldarchaeol production is high due to winter lake water sub-oxia. In lakes N3, Pluto, Rosaea, and Marshall brGDGT Illa" (Weber et al., 2015) is below detection limit in all samples, except one sample during the Early Holocene and four sample from the Middle Holocene in Lake Rosaea. In Lake Bullet, brGDGT Illa" concentration was below detection limit prior to 7 ka. After 7 ka, concentration increases, reaching a maximum between 6 and 3 ka (supplementary fig. S10). Afterwards, concentration remained low. The Holocene trends between cald/cren and Illa" are decoupled with each other, while trend between Illa" and HP5 and %Illa are similar, support our inferences of different production seasonality of GDGTs."

**Summary**

A more thorough consideration that lake-specific parameters, including mixing status and oxygen, contribute to brGDGT trends is warranted. Given that the key assumptions that lead to the conclusion that brGDGT distributions are entirely driven by IFS LWT are poorly supported, the discussion of climatic forcings that can explain the heterogeneous pattern of warming is somewhat moot and therefore not extensively evaluated at this stage of review. From this data, can we really be sure that the HTM occurs from ~7-5ka in this sector of Greenland with leads/lags around this timing related to continental position (with even this inconsistent across their dataset), or is it equally or more plausible that higher seasonality and warmer summers in the early Holocene led to mixing regime changes at some of these sites (reflected currently in isoGDGT0 concentration and 0/cren ratios) and consequently increased production of e.g., brGDGT Illa, creating a cold-biased temperature reconstruction in the early-middle Holocene and catchment-scale heterogeneity across the Holocene.

Reply: Thank you very much for these suggestions. In all the studied lakes, the cald/cren ratio is high from 10 to 6 suggesting sub-oxic conditions, and afterward low and stable, suggesting oxic conditions. In contrast, at most of the sites, fractional abundance of brGDGT Illa is high from 10 to 9 ka and from 5 ka to present, but low in between. If suboxic conditions inferred from cald/cren impacted the brGDGTs, we would anticipate a higher fractional abundance of brGDGT Illa from 10 to 6 ka. However, the observed fractional abundance of brGDGT Illa is not correlated with the cald/cren ratio in all lakes, except Rosaea, where cald/cren remains <45 for all but 2 samples. Therefore, we are

confident that the brGDGTs are not influenced by an oxygen-driven apparent cold-biased in the early to middle Holocene, and that the southwestern Greenland temperature maximum during the past 10 ka, occurred from 7-5 ka.

Please refer to the reply to the comments above for revised text.

**Other points of consideration:**

The cut-off of the IR6ME ratio at 0.3 to exclude data is arbitrary and substantially lower than the datasets that provide the reasoning for cut-off (i.e., 0.5, 0.4; Bauersachs et al., 2024; Novak et al., 2025). All data, including those flagged by IR ratio, should be included in the main figure(s), even if distinct symbols/colors are used to flag those datapoints, and a thorough discussion related to the IR ratio needs to be included. How does the data interpretation change if a cut-off of 0.4 or 0.5 is applied? Is this cut-off even still applicable when not using the modified MBT5Me calibration that excludes data above the cut-off?

Reply: We will apply the cut-off of the  $IR_{6Me}$  ratio at 0.4, similar to previous study (Novak et al., 2025). The cut-off of the  $IR_{6Me}$  ratio at 0.4 removes two samples from the Early Holocene in Lake Pluto and a bottom sample in Lake Bullet, which show abrupt (>+3) warm temperatures than preceding samples. We will remove these samples from temperature reconstructions, as non-thermal influence. We will up the text as:

"The isomer ratio of 5- to 6-methyl brGDGTs ( $IR_{6Me}$ ) ranges from 0.13 to 0.51 across all study lakes (supplementary fig. 6–10). Previously, a non-thermal effect on lacustrine brGDGTs has been identified when  $IR_{6Me}$  is greater than 0.4 (Bauersachs et al., 2023; Novak et al., 2025). In Lake Pluto two samples during the Early Holocene and in Lake Bullet bottom sample have  $IR_{6Me}$  greater than 0.4. Consequently, these samples were excluded from temperature reconstructions, considering non-thermal influence (Fig. 2). For Lake Gus, we employed the same screening for soil-derived brGDGTs (%Hexamethylated) as used in Cluett et al., (2023)"

The authors compare IFS LWT directly to estimates of air temperature (e.g., from ice cores, chironomid, pollen), which based on their own model, aren't directly comparable. The GISP2 record that is presented is not corrected for elevation change or seasonal bias (see Axford et al., 2021). The magnitude of warming in these data is much higher compared to existing estimates of temperature change, and this discrepancy warrants discussion and reconciliation with both existing temperature reconstructions and the model that suggests lake water response should be dampened compared to air temperature if these data are interpreted as temperature.

Reply: We agree that direct comparison between IFS LWT and air temperature reconstructions from proxies such as ice cores, chironomids, and pollen can be problematic. Now, we will inferred mean air temperature for the months above freezing based average of three air temperature calibrations in the region (Otiniano et al., 2023, 2024; Raberg et al., 2021). We will update the entire manuscript accordingly.

We note that GISP2 record is influenced by elevation changes as:

"However, it should be noted that the temperature timeseries from GISP2 is influenced by Holocene ice sheet elevation change (Axford et al., 2021; Martin et al., 2024)."

More justification is needed for the selection of a site-specific calibration dataset from one lake in south Greenland (Zhao et al., 2021) vs. calibration datasets that incorporate more extensive data and many additional Arctic sites (e.g., Raberg et al., 2021) that have greater potential to more adequately cover the range of environmental conditions that occur in these 7 lakes through time.

Reply: We will add a more detailed discussion for the reasoning behind the choice of brGDGTs to temperature calibration. Now, we will inferred mean air temperature for the months above freezing based average of three air temperature calibrations in the region (Otiniano et al., 2023, 2024; Raberg et al., 2021). We will update section 3.1 as:

"While there are several lacustrine brGDGT-temperature calibrations available globally (Martínez-Sosa et al., 2021; Raberg et al., 2021; Zhao et al., 2023), regionally (Bauersachs et al., 2023; Dang et al., 2018; Otiniano et al., 2023, 2024; Russell et al., 2018), and site-specifically (Zhao et al., 2021; Bittner et al., 2022; Feng et al., 2019), only Raberg et al. (2021), Zhao et al. (2021), and Otiniano et al. (2023; 2024) are based on the HPLC method separating GDGTs isomers and have considered the Arctic seasonal climate. Therefore, we assess these calibrations to infer temperature in the southwestern Greenland lakes. The Raberg et al. (2021) and Otiniano et al. (2023; 2024) calibrations estimate mean air temperature for the months above freezing (MAF) while Zhao et al. (2021) estimate epilimnion lake water temperature for the ice-free season (Ice free season lake water temperature, IFS LWT). For all analyzed samples, the reconstructed MAF based on Raberg et al. (2021) yields a temperature range from ~3.9 to 11.6 °C; using Otiniano et al. (2024) yields a temperature range from 3.8 to 8.6 °C; and using Otiniano et al. (2023) yields a temperature range from 5.8 to 9.9 °C, while IFS LWT reconstrued using Zhao et al. (2021) yields a temperature range from ~5.9 to 21.6 °C (supplementary fig. S14). For lakes Marshall, N3, Pluto and Rosaea, all four calibrations generated similar temperature trends throughout the Holocene, while the absolute values are different. For Bullet Lake, the MBT"5ME-based calibrations (Otiniano et al. (2024), Otiniano et al. (2023), and Zhao et al. (2021)) yielded similar trends, whereas Raberg et al. (2021) yielded a divergent trend, albeit with all the calibrations providing different absolute values.

In Arctic lakes, the water temperature change in step with air temperature but the magnitude of the water temperature response is smaller and delayed compared to air temperature (Cluett et al., 2023; Kettle et al., 2004; Piccolroaz et al., 2020). The lake model sensitivity tests that we conduct here suggest a lake temperature response of ~0.4 to 0.6°C for every 1°C annual or JJAS air temperature change (Fig. 5 and supplementary fig. S16). Considering this relationship between air and lake temperature responses, Zhao et al. (2021) based brGDGTs to IFS LWT provides an excessively high estimation of the air temperature change during the Holocene, ranging from +10 to +15 degrees. The estimation of MAF based on Otiniano et al. (2024; 2023) and Raberg et al. (2021) are more reasonable for southwestern Greenland. Therefore, we used the average of these three calibrations to estimate the MAF in our study (Otiniano et al., 2023, 2024; Raberg

et al., 2021). We estimate the error as the standard deviation between these three calibrations."

Further, we will update the entire manuscript to mean air temperature for months above freezing (MAF).

Is the same screening for soil-derived brGDGTs (<30% hexa-methylated) used in Cluett et al. for the Lake Gus temperature reconstruction also applied to all other lake brGDGT data presented here?

Reply: No, the same screening for soil-derived brGDGTs (<30% hexa-methylated) used in Cluett et al. (2023) for the Lake Gus temperature reconstruction was not applied to all other lake brGDGT data presented here. Instead, we applied the IR6ME cutoff value, as discussed above.

Why does the lake model for Lake Gus in Cluett et al. have such different sensitivity between air and lake water temperature compared to the model for the other lakes here? It could be valuable to remodel Lake Gus and model Lake 578 here as well, using the same modeling decisions applied to the other lakes.0

Reply: There is no major difference in sensitivity between air and lake water temperature in Lake Gus compared to the other lakes. The sensitivity test results for Lake Gus, as presented in Cluett et al., 2023, are shown in absolute values, whereas we presented results as anomalies relative to modern values.

We reran the lake model for Lake Gus using the lake parameters suggested by Cluett et al., (2023), and obtained similar results to both Cluett et al (2023) and to lakes Bullet, Marshall, Rosaea, and Pluto, which have similar morphometry. We also ran the lake model for Lake 578 using the same lake parameters as for Cluett et al (2023), given its similar morphometry.

An internal C46 standard is mentioned as added but it is not elaborated on if or how this standard was used in data processing

Reply: We will add a sentence has been to the text clarifying the role of the internal standard C46 and how it was applied during data processing, as follows: "We determined the GDGT concentrations in relation to the C46 internal standard."

**Presentation Quality**

The structure of this manuscript overall flows well. At times the presentation is a little odd though, for example, the supplemental figures are referenced well-before and more often than most of the main figures. It seems like at least some of these data should be moved into the main text (e.g., Figs. S6-10). Some of the background/discussion is not internally consistent. There are minor issues with typos (e.g., surface areas in Table 1 are different from surface areas given in the in-line text; Comarum Sø is spelled incorrectly in parts of the text, Caldarchaeol is spelled incorrectly in some of the figure captions and I did wonder

why it's presented as "Cald" as opposed to the more common presentation of isoGDGT0, etc.). The scaling of deg C on the y-axes on Figures 2, 3 are confusingly unique by each row and make it hard to compare temperature change site to site and record to record; I noted a similar issue with the scaling of cald/cren ratios across Figs. S6-10.

Reply: We refer to supplementary figures after the main figure in the revised paper, whenever possible.

We will add figure 2 showing the ratio of cald/cren, Hexa%, and MBT'5ME from all study lakes in the main text. Further, we believe that moving entire supplementary figs S6 to S10 to main text will not provide additional information to the readers of this paper. Therefore, we prefer to keep them in supplement as they are now.

We will carefully check the paper and made the necessary changes to make the background/discussion internally consistent.

We will double-check the surface area of the lake and will correct, if needed. The paper is carefully proofread for spelling errors.

Caldarchaeol is the scientific name for isoGDGT-0. We have used Caldarchaeol in our group's previous publications (Cluett et al., 2023; Holtzman et al., 2025). To maintain consistency, we would like to use Caldarchaeol in this publication as well.

We will revise the y-axis scales for temperature (Figures 2 and 3) and cald/cren ratios (Figures S6–10) to ensure consistency across sites, improving comparability between the records.

[revised manuscript text omitted]